

# Acquisition of fungi from the environment modifies ambrosia beetle mycobiome during invasion

Davide Rassati[1], Lorenzo Marini[1] and Antonino Malacrinò[2]

[1] Department of Agronomy, Food, Natural Resources, Animals and Environment (DAFNAE), University of Padova, Padova, Italy
[2] Department of Evolution, Ecology and Organismal Biology, Ohio State University, Columbus, OH, United States of America

## ABSTRACT

Microbial symbionts can play critical roles when their host attempts to colonize a new habitat. The lack of symbiont adaptation can in fact hinder the invasion process of their host. This scenario could change if the exotic species are able to acquire microorganisms from the invaded environment. Understanding the ecological factors that influence the take-up of new microorganisms is thus essential to clarify the mechanisms behind biological invasions. In this study, we tested whether different forest habitats influence the structure of the fungal communities associated with ambrosia beetles. We collected individuals of the most widespread exotic (*Xylosandrus germanus*) and native (*Xyleborinus saxesenii*) ambrosia beetle species in Europe in several old-growth and restored forests. We characterized the fungal communities associated with both species via metabarcoding. We showed that forest habitat shaped the community of fungi associated with both species, but the effect was stronger for the exotic *X. germanus*. Our results support the hypothesis that the direct contact with the mycobiome of the invaded environment might lead an exotic species to acquire native fungi. This process is likely favored by the occurrence of a bottleneck effect at the mycobiome level and/or the disruption of the mechanisms sustaining co-evolved insect-fungi symbiosis. Our study contributes to the understanding of the factors affecting insect-microbes interactions, helping to clarify the mechanisms behind biological invasions.

## INTRODUCTION

Insect invasions represent one of the most demanding challenges today (*Leemans & De Groot, 2003*). Preventive measures adopted so far (*Ormsby & Brenton-Rule, 2017*) have slowed down but not stopped these events (*Haack et al., 2014*), and further invasions are expected to occur (*Seebens et al., 2017*). One reason for the limited efficacy of existing biosecurity systems is the still overlooked role of microorganisms in invasion ecology (*Lu, Hulcr & Sun, 2016*; *Amsellem et al., 2017*; *Linnakoski & Forbes, 2019*). Insects, like many other organisms, live in association with bacterial and fungal symbionts (*Douglas, 2015*; *Gurung, Wertheim & Falcao Salles, 2019*), which can have a positive (i.e., mutualistic),

Corresponding authors
Davide Rassati,
davide.rassati@unipd.it
Antonino Malacrinò,
malacrino.1@osu.edu,
antonino.malacrino@gmail.com

negative (i.e., parasitic) or neutral (i.e., commensalistic) impact on their host's fitness. These symbionts can also facilitate (*Lu et al., 2010*; *Himler et al., 2011*; *Adams et al., 2011*; *Vilcinskas et al., 2013*) or limit (*Zhou et al., 2018*; *Umeda & Paine, 2019*) the invasion process of their insect host. When invading a new environment, insects and their microorganisms experience biotic and abiotic forces that can lead to the loss of part of the microbiome (*Lester et al., 2017*). This "bottleneck effect" may predispose exotic insects to acquire microorganisms from the invaded environment (*Hajek et al., 2013*; *Wooding et al., 2013*; *Taerum et al., 2013*; *Wingfield et al., 2017*). These microorganisms may confer important ecological adaptations, such as heat tolerance or parasite defense, influencing insects' ability to establish and spread in the invaded environment (*Oliver et al., 2010*; *Henry et al., 2013*). Clarifying the ecological factors and dynamics behind the acquisition of microorganisms during insect invasions is an essential step to plan effective biosecurity programs.

One of the most complex examples of symbiosis in forest ecosystems occurs between wood-boring ambrosia beetles (Coleoptera; Scolytinae and Platypodinae) and ambrosia fungi (Ascomycota: Microascales, Ophiostomatales) (*Hulcr & Stelinski, 2017*; *Vanderpool, Bracewell & McCutcheon, 2018*). Adult females acquire mutualistic ambrosia fungi from the parental nest and transport them to newly established nests inside specific organs (i.e., mycetangia) or inside their guts (*Francke-Grosmann, 1963*; *Francke-Grosmann, 1967*). Then, beetles farm the fungi within the wood galleries they live in (*Biedermann & Taborsky, 2011*), and feed on them as both larvae and adults (*Batra, 1966*). Besides these obligate nutritional mutualists, ambrosia beetles carry several other fungal symbionts in both the mycetangium and other body parts (*Kostovcik et al., 2015*; *Freeman et al., 2016*; *Bateman et al., 2016*; *Malacrinò et al., 2017*; *Miller et al., 2019*). These can be commensals, parasites or facultative mutualists (*Skelton et al., 2018*). The complexity of these symbioses is still largely unresolved, in particular considering the potential interactions among exotic insects and native fungi occurring in the invaded environment.

Several ambrosia beetle species have successfully established outside their native range in the last two decades (*Rassati, Lieutier & Faccoli, 2016*; *Rabaglia et al., 2019*). Nonetheless, the spread of several species has been limited by climatic conditions (e.g., humidity, temperature). Exotic ambrosia beetles are indeed able to survive only in areas suitable for the growth of their fungal symbionts (*Marini et al., 2011*; *Rassati et al., 2016a*; *Rassati et al., 2016b*; *Zhou et al., 2018*; *Umeda & Paine, 2019*). This scenario could however change if an exotic beetle is able to acquire native fungi from the invaded environment. This acquisition can occur through (i) the exchange of fungi between native and exotic species, and/or (ii) the direct contact with the mycobiome of the invaded environment. The exchange of fungi between native and exotic ambrosia beetles can occur between two species with neighboring galleries, when fungi grow from the gallery of one species to that of the other (*Carrillo et al., 2014*). This mechanism is expected to involve primary or facultative mutualists and may not be unusual, particularly because different species of ambrosia beetles select their host plant in a similar way, so different species may colonize the same tree (*Ranger et al., 2015*). The second mechanism, instead, may occur when adult females searching for a new host come in contact with native fungi present in the environment (*Seibold et al., 2019*).

This mechanism should involve fungi that mainly establish commensalistic relationships with the beetles, but plant pathogens can also be involved (*Juzwik et al., 2016*; *Ploetz et al., 2017*; *Chahal et al., 2019*). Currently, the frequency and the extent of these associations is largely unclear.

In this study, we tested the hypothesis that different habitats influence the composition of the fungal community associated with ambrosia beetles, reflecting a potential acquisition of fungi from the environment. We used a metabarcoding approach for identification of the fungal community of the most widespread exotic (*Xylosandrus germanus*) and native (*Xyleborinus saxesenii*) ambrosia beetle species in European forests. Individuals of both species were collected in two forest habitats: old-growth forests and restored forests. Old-growth forests are expected to host more complex fungal communities than restored forests (*Blaser et al., 2013*; *Pioli et al., 2018*); thus, we hypothesized that ambrosia beetles should reflect these differences in their mycobiome. Furthermore, the mechanisms regulating insect-fungus symbioses resulting from a long co-evolutionary history (*Biedermann, De Fine Licht & Rohlfs, 2019*) might be disrupted by the interaction with microbiomes of the invaded habitat. Therefore, when comparing the fungal communities associated to individuals collected in the two forest habitats, we expected to observe larger differences for the exotic than for the native ambrosia beetle species.

## MATERIALS & METHODS

### Ambrosia beetle species

We selected two ambrosia beetle species: the exotic *X. germanus* and the native *X. saxesenii*. *Xylosandrus germanus* is a species native to Asia that was first reported in Europe in the 1950s and since then rapidly spread, becoming one of the dominant ambrosia beetles in European forest ecosystems (*Galko et al., 2018*). *Xylosandrus germanus'* fungal mutualist is *Ambrosiella grosmanniae* (*Mayers et al., 2015*). *Xyleborinus saxesenii* is instead a species of Palaearctic origin and its main fungal mutualist is *Raffaelea sulfurea*, although other fungi have been found in association with this beetle species (*Biedermann et al., 2013*).

### Sampling locations and procedure

Beetles were collected in 2016 in ten forest stands located in the Northeast of Italy (Fig. S1 and Table S1), across two forest habitats: old-growth forests ($n = 5$) and restored forests ($n = 5$). With "old-growth forests", we refer to the remnants of the old oak–hop-hornbeam forest (*Quercus* spp. and *Ostrya carpinifolia* Scop.) that covered the vast majority of Veneto and Friuli Venezia Giulia regions after the last ice age. With "restored forests", we refer to mixed forests that were planted over the last 30 years to restore forests across agricultural landscapes. Both forest habitats are dominated by oak (*Quercus* spp.), ash (*Fraxinus* spp.), maple (*Acer* spp.), and hop-hornbeam (*O. carpinifolia*). In addition, both forest habitats are present in relatively small patches embedded in an agriculture-dominated landscape (min = 2.65 ha, max = 165.15 ha for old-growth forests; min = 2.37 ha, max = 37.41 ha for restored forests).

Beetles were trapped using green and purple 12-multi-funnel traps (Synergy Semiochemicals, Burnaby, Canada) baited with ultra-high release rate ethanol pouches

(99% purity, release rate of 300–400 mg/day at 20 °C, Contech Enterprises). Although ethanol is attractive for a wide range of wood-borers (*Miller, 2006*), it is also the most commonly used volatile for trapping ambrosia beetles (*Reding et al., 2011*). The ethanol pouch was always attached to the sixth funnel and hung outside the trap. Traps were set in the understory at about 1.5 m above the ground and were suspended at least 1m from the tree bole. Trap collecting cups were half-filled with 1:1 solution (v/v) of ethylene glycol:water to kill and preserve captured beetles (known as "wet system") (*Steininger et al., 2015*). At each trap check, collecting cups were emptied and the solution was renewed. Traps were set up in mid-May and emptied every three weeks until the beginning of August. At each visit, all insects were collected, put in tubes filled with ethanol, and brought to the laboratory where ambrosia beetles were separated from other trapped insects. Each individual was then morphologically identified to species level and kept in separate vials filled with ethanol until they were processed. Then, we retained for the analysis only individuals of the two ambrosia beetle species that were simultaneously collected during the same trapping period and in the same trap. This allowed an intra-trap comparison to test for cross-contamination between beetle species (see 'Results', Table S2). Our sampling procedure did create the possibility of microbial cross-contamination among the different insect specimens simultaneously present in the trap collector cup (*Viiri, 1997*). However, in our previous work we demonstrated that individuals collected in the same trap do not show evidence of cross-contamination (*Malacrinò et al., 2017*). In an effort to reduce possible environmental contamination, we also sterilized the external surface of the insect body. First, we put each insect in a vial with ddH$_2$O in a water bath and sonicated them for 1 min. After sonication, we washed each insect by vortexing once in ethanol (100%), twice in sodium hypochlorite (5%), and twice in ddH$_2$O for 1 min following each wash step. For each ambrosia beetle species, we processed 15 individuals per sampling site (total of 300 individuals).

## DNA extractions, libraries preparation and amplicon sequencing

Single individuals were crushed in an extraction buffer (10 mM Tris, 100 mM NaCl, 10 mM EDTA, 0.5% SDS) using three one mm ∅ stainless steel beads per tube, with the aid of a bead mill homogenizer set at 30 Hz for 5 min (TissueLyzer II, Qiagen, UK). The mixture was treated with proteinase K (5Prime GmbH, Germany) following the producer's instructions. Total DNA was extracted using the MoBio PowerSoil Kit (Mo Bio Laboratories, Inc., Carlsbad, CA, USA) following the manufacturer's protocol. DNA concentration and purity were assessed with a Nanodrop 2000 spectrophotometer (Thermo Fisher Scientific Inc., USA).

The fungal community associated with each individual was characterized by amplicon sequencing targeting the ITS2 region using gITS7 and ITS4 primers, as previously indicated by *Kostovcik et al. (2015)*. We selected the ITS2 region (*Nilsson et al., 2019*) to ensure we captured the diversity of fungi with which beetles come in contact during host searching. We are aware that the ITS2 region can lead to an amplification bias for Microascales and Ophiostomatales (*Kostovcik et al., 2015*), the two orders including the main mutualists of *X. germanus* and *X. saxesenii*. Here, however, we are interested in the entire mycobiota,

which has been less frequently described and might explain important aspects of beetle ecology. PCR reactions were performed in a total volume of 25 µl, containing about 50 ng of DNA, 0.5 µM of each primer, 1X KAPA HiFi HotStart ReadyMix (KAPA Biosystems, USA) and nuclease-free water. Amplifications were performed in a Mastercycler Ep Gradient S (Eppendorf, Germany) set at 95 °C for 3 min, 98 °C for 30 s, 56 °C for 30 s and 72 °C for 30 s, repeated 30 times, and ended with 10 min of extension at 72 °C. Each amplification was carried out in technical triplicate, including three non-template controls with nuclease-free water instead of DNA. Nuclease-free water (100 µl) was also processed using the same procedure as the experimental samples (from DNA extraction to sequencing) in order to exclude contamination of reagents and instruments. Amplification success was checked by electrophoresis on 1.5% agarose gel stained with GelRed (Biotium Inc., Fremont, CA, USA). Although we did not observe any amplification bands for the negative/non-template control samples, these were processed and sequenced together with experimental samples. PCR products from the same sample were then pooled together, and cleaned using Agencourt AMPure XP kit (Beckman Coulter, Brea, CA, USA) following the producer's instructions. A further short-run PCR was performed to integrate Illumina i7 and i5 indexes following the producer's protocol (Nextera XT, Illumina, San Diego, CA, USA), and amplicons were purified again as explained above. Libraries were then quantified with the Invitrogen Qubit HS dsDNA kit (Invitrogen, Carlsbad, CA, USA), normalized to a concentration of 10 ng/µl using nuclease-free water, pooled together and sequenced with an Illumina MiSeq sequencer, using the MiSeq Reagent Kit v3 300PE chemistry (Illumina, San Diego, CA, USA) following the producer's protocol.

## Data analysis

Demultiplexed paired-end reads were merged using the PEAR 0.9.1 algorithm using default parameters (*Zhang et al., 2014*). Raw data handling was carried out using QIIME 1.9 (*Caporaso et al., 2012*), quality filtering reads with default parameters, binning Operational Taxonomic Units (OTUs) using open-reference OTU-picking through UCLUST algorithm (97% similarity), and discarding chimeric sequences discovered with USEARCH 6.1 (*Edgar, 2010*). All non-fungal OTUs were discarded using ITSx (*Bengtsson-Palme et al., 2013*). Taxonomy assignment was performed using the BLAST method (default parameters) by querying towards a custom database built using all ITS2 reference sequences deposited at NCBI GenBank (accessed on July 2017). R statistical environment v3.5.1 (*R Core Team, 2013*) plugged with the packages *vegan* (*Dixon, 2003*), *phyloseq* (*McMurdie & Holmes, 2013*), *picante* (*Kembel et al., 2010*) and *DESeq2* (*Love, Huber & Anders, 2014*) was used for data analysis. First, singletons and samples with fewer than 1,000 counts were removed. Data processing resulted in a dataset of 3,634,647 reads clustered into 19,744 OTUs. Then, comparisons of fungal community composition between ambrosia beetle species and between forest habitats within the same beetle species were performed using PERMANOVA analysis (999 permutations stratified at site level) calculated on a UniFrac distance matrix. Non-metric multidimensional scaling (NMDS) procedure was performed to visualize differences in the structure of fungal communities. The diversity of fungal communities was assessed using Chao1 (total diversity) (*Chao, 1984*), Faith's phylogenetic
diversity (which considers both total diversity and the phylogenetic relationship between taxa within the community) (*Faith, 1992*), and 1-Simpson (dominance) (*Simpson, 1949*) indexes. Comparisons were performed using mixed-effects models (one model for each diversity index) with the *lmer* function under the *lme4* R package (*Bates et al., 2015*) using ambrosia beetle species and forest habitat as factors, and sampling site as a random variable. The package *emmeans* was used to infer pairwise contrasts within mixed-effects models (FDR corrected). The use of "sampling site" for stratification in PERMANOVA and as a random variable in the mixed-effects model allowed the control of both non-homogeneity in sample number at each trap (Table S2) and potential spatial effects. The differential presence of OTUs between forest habitats and within the same beetle species was assessed using the package *DESeq2*, by contrasting the two forest types within each species. The association of each fungal genus to a functional guild was performed by searching against the FUNGuild database (*Nguyen et al., 2016*) and manually curating the results in case of multiple results from the same query.

## RESULTS

### Fungal communities associated with *X. germanus* and *X. saxesenii*

The reconstruction of the fungal communities showed that the exotic ambrosia beetle *X. germanus* and the native ambrosia beetle *X. saxesenii* are associated with different fungi ($F_{1,211} = 10.5$; $P < 0.001$). The absence of cross-contamination was shown by a multiple comparison procedure following PERMANOVA: differences between ambrosia beetle species were found at all sites ($P < 0.01$ FDR corrected–Table S2). In case of cross-contamination we would expect an overlap of the fungal communities and, thus, no differences.

The fungal communities of both ambrosia beetle species were dominated by unidentified taxa (83.97% in *X. germanus* and 73.91% in *X. saxesenii*–Table S3). Instead, we identified 26 genera that include plant pathogens (4.75% in *X. germanus* and 5% in *X. saxesenii*) mainly represented by the genus *Cladosporium*. The rest of the communities were represented by saprotrophs (2.73% in *X. germanus* and 4.12% in *X. saxesenii*), yeasts (5.35% in *X. germanus* and 15.96% in *X. saxesenii*) (Table S3) and at low relative abundances insect pathogens, mycorrhizal fungi, endophytes and lichen parasites (Table S3). In addition, in both ambrosia beetle species we found sequences that can likely be assigned to the respective main mutualists: *Ambrosiella* sp. in *X. germanus* (2.25%) and *Raffaelea* sp. in *X. saxesenii* (0.01%).

### Effect of forest habitat on fungal communities

Using a PERMANOVA analysis we found that the fungal community segregated by forest habitat both in the exotic *X. germanus* ($F_{1,10} = 21.8$; $P < 0.001$—Fig. 1A) and in the native *X. saxesenii* ($F_{1,10} = 1.6$; $P = 0.004$—Fig. 1B), although the effect was much more evident in *X. germanus* ($R^2 = 0.16$) than in *X. saxesenii* ($R^2 = 0.01$). A different pattern between *X. germanus* and *X. saxesenii* also emerged when looking at the diversity of the fungal communities. For *X. germanus*, individuals collected in old-growth forests were associated with a richer and more diverse fungal community than those collected in restored forests

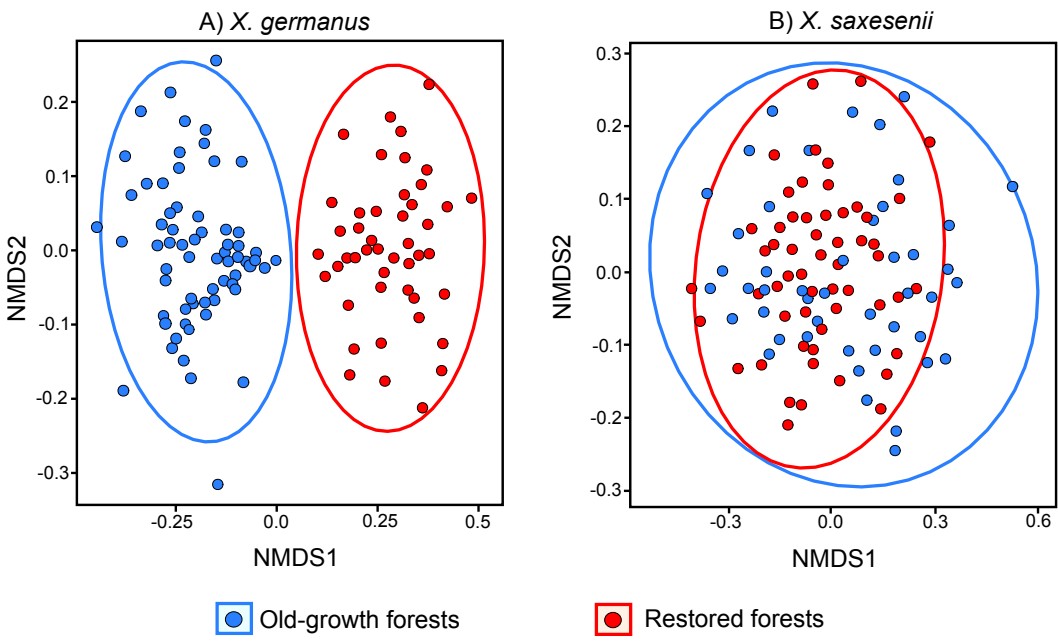

**Figure 1** NMDS (Non-metric Multi Dimensional Scaling) analysis of the fungal communities associated with the exotic ambrosia beetle *X. germanus* (A) and the native ambrosia beetle *X. saxesenii* (B) in old-growth forests and restored forests.

($P < 0.001$ for both Chao1 and phylogenetic diversity—Figs. 2A and 2B, Table S4), whereas these differences were not observed in the native *X. saxesenii* ($P > 0.05$ for both Chao1 and phylogenetic diversity—Figs. 2A and 2B, Table S4). On the contrary, in both *X. germanus* and *X. saxesenii* the dominance index (1-Simpson) significantly differed between the two forest habitats ($P < 0.001$—Fig. 2C, Table S4). In particular, we observed a higher dominance index in restored forests compared to old-growth forests for *X. germanus*, and a higher dominance index in old-growth forests versus restored forests for *X. saxesenii*.

Comparing the fungal community associated with individuals collected in the two forest habitats, for *X. germanus* we found 121 differentially abundant OTUs: 4 of them were more abundant in restored than in old-growth forests (1 *Ambrosiella* sp., 1 *Aspergillus* sp., 1 *Saccharomyces* sp. and 1 unidentified, Fig. 3A), whereas 117 were more abundant in old-growth than in restored forests (102 unidentified OTUs, and 15 genera, Fig. 3A). The same analysis on *X. saxesenii* resulted in 4 differentially abundant OTUs (1 *Aureobasidium* sp. and 3 unidentified), all of them more abundant in restored than in old-growth forests (Fig. 3B).

## DISCUSSION

Absence of adaptation, or low plasticity, in the microbiota of an exotic species can limit its establishment in a new environment (*Rassati et al., 2016b*; *Umeda & Paine, 2019*). The acquisition of microorganisms native to the invaded environment may however help the exotic species to overcome these ecological barriers. Yet, this topic is still in its

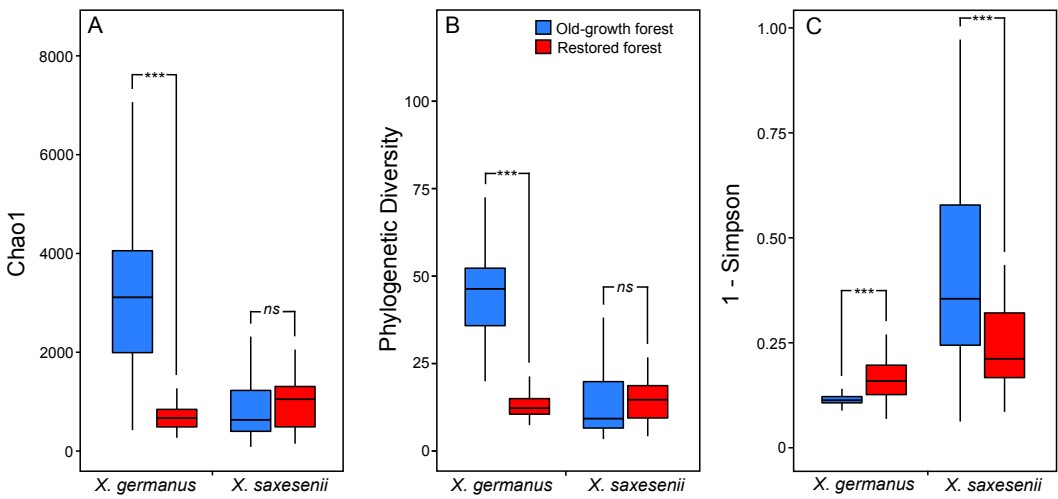

**Figure 2** Alpha-diversity indices (A, Chao 1; B, Faith's Phylogenetic Diversity; and (C) 1-Simpson) for fungal communities associated with the exotic ambrosia beetle *X. germanus* and the native ambrosia beetle *X. saxesenii* in old-growth and restored forests. $***=P<0.001$; $ns=P>0.05$. Full results from mixed-effect models are reported in Table S4.

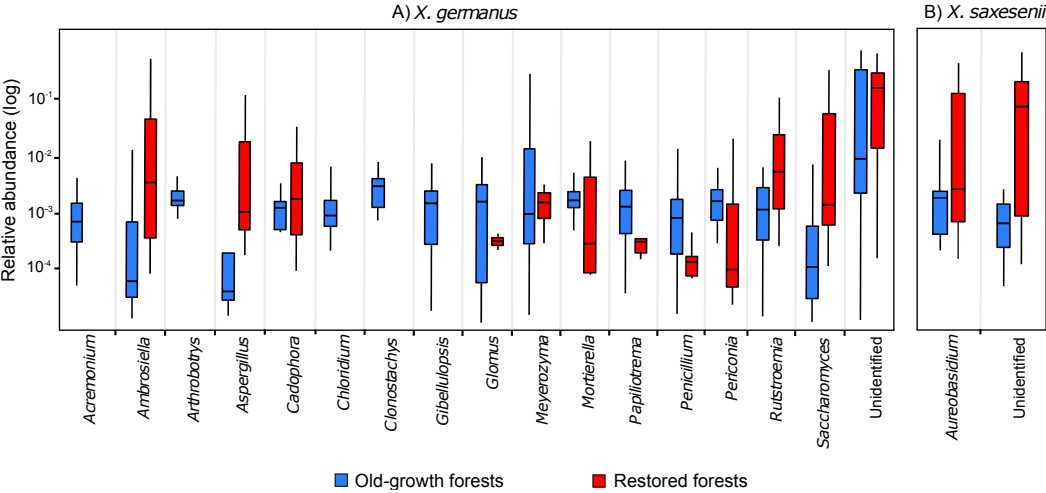

**Figure 3** Relative abundance (log-scale) of fungal genera in the exotic ambrosia beetle *X. germanus* (A) and the native ambrosia beetle *X. saxesenii* (B) associated to individuals collected in old-growth forests vs restored forests. Genera represented in this figure are those that resulted to be significantly differentially abundant between the two forest habitats (cutoff $P=0.05$ FDR corrected). Each genus is represented by a single OTU (excluding *Glomus* and *Mortierella*, each with 2 OTUs), while the category 'Unidentified' is represented by 103 OTUs in *X. germanus* and 3 OTU in *X. saxesenii*.

infancy and the mechanisms leading to the acquisition of new microorganisms are still understudied. We found that forest habitat shaped the mycobiome associated with the exotic ambrosia beetle *X. germanus*, potentially reflecting the acquisition of fungi from the invaded environment. In addition, we showed a stronger effect of forest habitat on the

fungal community associated with the exotic *X. germanus* compared to the native ambrosia beetle *X. saxesenii*. This suggests that two (non-mutually exclusive) mechanisms may have occurred: (i) a bottleneck effect that caused the loss of part of the original microorganisms; and (ii) the disruption of the mechanisms sustaining co-evolved insect-fungi symbiosis.

In our study, the exotic ambrosia beetle *X. germanus* and the native *X. saxesenii* were associated with different fungal communities. Although both species are highly polyphagous and have overlapping phenology, they can show different preferences in host tree species (*Rassati et al., 2016a*), ethanol content in host tissues (*Rassati et al., 2019*) or vertical strata (*Menocal et al., 2018*), which could lead to interactions with different fungal communities. We were unable to taxonomically identify the majority of OTUs due to the lack of reliable taxonomic information (*Stielow et al., 2015*; *Abdelfattah et al., 2018*). Among the identified taxa, however, we found a large cohort of plant pathogens, saprotrophs and yeasts, of which many have already been reported to establish a commensalistic relationship with both bark and ambrosia beetles (*Kostovcik et al., 2015*; *Davis, 2015*; *Miller et al., 2016*; *Malacrinò et al., 2017*).

We found that forest habitat greatly influenced the diversity and dominance of fungal communities associated with the exotic ambrosia beetle *X. germanus*. A similar pattern was previously shown only for the invasive ambrosia beetle *Xyleborus glabratus*, where sampling location influenced the structure of the symbiotic fungal community (*Campbell et al., 2016*). Here, we show that individuals of the exotic *X. germanus* were associated with a richer, more diverse and more even community of fungi in old-growth forests than in restored forests. This pattern reflects the different fungal community structures likely inhabiting the two forest habitats and suggests the occurrence of a direct acquisition of fungi from the environment during invasion. Future research efforts should directly compare the mycobiome associated with ambrosia beetles to the environmental fungal communities, proving empirical evidence that such acquisition occurs. After introduction in a new environment, an exotic insect and its microbiome experience a series of biotic and abiotic forces that may lead the insect to lose part of its original community of microorganisms. This "*bottleneck effect*" challenging the microbiome (e.g., *Lester et al., 2017*) may favor the acquisition of microorganisms from the invaded habitat. Given that we do not have data on the community of fungi associated with *X. germanus* in its native area, we cannot state whether a bottleneck effect occurred. Along with the depletion of the original mycobiome, we speculate that an exotic species may be prone to acquire new microorganisms due to the potential mismatch of the mechanisms maintaining symbioses with the invaded ecosystems. Symbioses are the result of a long co-evolution, and both the host and the symbionts present a series of chemical, structural, and genomic co-adaptations (*Blaz et al., 2018*; *Mayers et al., 2019*; *Skelton et al., 2019*; *Biedermann, De Fine Licht & Rohlfs, 2019*; *Veselská et al., 2019*). The mechanisms that serve to maintain existing symbiosis may be challenged by the newly encountered microbiomes and might not work properly, leading to the establishment of new associations. While most of the fungi may represent transient associations, it is possible that some can compete for resources with the primary mutualists present in the mycentangium (*Castrillo, Griggs & Vandenberg, 2016*; *Menocal et al., 2017*). Whether such a mechanism occurred, however, cannot be stated. Indeed, by analyzing

the whole insect, we are not able to determine if the fungal taxa we identified inhabited the mycetangium or the insect's guts. This is an important aspect to investigate in future studies as a switch in the fungal symbionts in the mycentangium may lead to important consequences for the beetle fitness (*Skelton et al., 2019*).

We also found a weak environmental effect on the native species. The microbiome of native insects have been shown to vary with habitat (*Yun et al., 2014*; *Kudo et al., 2019*), thus we expected some differences among the fungal communities associated with *X. saxesenii* individuals collected in the different forest habitats. In our study, however, differences were very small compared to those observed for *X. germanus*, and were found only in terms of dominance. Specifically, the community of fungi associated with *X. saxesenii* was more even in restored than old-growth forests. This pattern can be explained by the different microclimatic conditions and nutrient availability present in the two forest habitats which may have favored certain fungi rather than others.

## CONCLUSIONS

A timely topic in invasion ecology is the understanding of the mechanisms by which exotic species establish novel symbiotic associations in the invaded environment (*Lu, Hulcr & Sun, 2016*; *Amsellem et al., 2017*). Despite we analyzed only one exotic ambrosia beetle species, our results support the hypothesis that the direct acquisition of microorganisms from the environment can modify the microbiome of an exotic species. Species distribution models are commonly used to plan invasive species surveillance programs and decide where to concentrate efforts and resources (*Lantschner, De la Vega & Corley, 2019*). These models are based on known occurrence records and the environmental conditions at occurrence localities to predict where a certain species can establish outside its native range. The acquisition of novel microorganisms in the invaded environment, however, may alter predictions for the establishment and spread of exotic species. Incorporating the role of microbes into ecological theories is thus fundamental to clarify the mechanisms behind insect invasions and aid in biosecurity surveillance.

## ACKNOWLEDGEMENTS

The authors thank Peter Biedermann, Alison Bennett, Philip Smith and the two anonymous reviewers for their insightful comments on an earlier draft of this manuscript, and Matteo Marchioro for field assistance.

### Funding

This study was supported by the Fund for Basic Research Activities (FBAR)–ANVUR –Italian National Agency for the Evaluation of the University and Research Systems, and European Union's Horizon 2020 research and innovation programme under grant agreement No 771271. The funders had no role in study design, data collection and analysis, decision to publish, or preparation of the manuscript.

## Competing Interests

The authors declare there are no competing interests.

## Author Contributions

- Davide Rassati conceived and designed the experiments, performed the experiments, contributed reagents/materials/analysis tools, prepared figures and/or tables, authored or reviewed drafts of the paper, approved the final draft.
- Lorenzo Marini conceived and designed the experiments, prepared figures and/or tables, authored or reviewed drafts of the paper, approved the final draft.
- Antonino Malacrinò conceived and designed the experiments, performed the experiments, analyzed the data, prepared figures and/or tables, authored or reviewed drafts of the paper, approved the final draft.

## Data Availability

Raw reads are available at NCBI SRA under the BioProject PRJNA524707.

## Supplemental Information

Supplemental information for this article can be found online at http://dx.doi.org/10.7717/peerj.8103#supplemental-information.

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
