# Peer review of "Acquisition of fungi from the environment modifies ambrosia beetle mycobiome during invasion"

_PeerJ, doi:10.7717/peerj.8103_

## Round 0.1 · original submission · Major Revisions

Dear Dr. Rassati and colleagues:

Thanks for submitting your manuscript to PeerJ. I have now received two independent reviews of your work, and as you will see, the reviewers raised some concerns about the research. Despite this, these reviewers are optimistic about your work and the potential impact it will have on research communities studying ambrosia beetle ecology and evolution. Thus, I encourage you to revise your manuscript, accordingly, taking into account all of the concerns raised by both reviewers.

Please improve the focus of the Discussion section, and also have an English expert review the manuscript for content, clarity and grammar. Please note that reviewer 2 provided a marked-up copy of your manuscript with many helpful suggestions.

I look forward to seeing your revision, and thanks again for submitting your work to PeerJ.

Good luck with your revision,

-joe

Reviewer 1 ·

Basic reporting

Basic Reporting

English needs to be checked and revised.
Examples of sentences with poor English
Lines:39-46 (see minor comments section)
Lines:90-91
Lines:71-72
Line: 89
Lines: 288-289
Lines:293
Lines:325
The current phrasing makes comprehension difficult in places.

Intro and background show the context well and the literature is well referenced and relevant.
However, I would like to see some discussion of literature relating to co-evolution between fungal symbionts and beetle hosts included as a key hypothesised implication of their results relate to
"a weaker co-evolutionary association in the invaded environment or the occurrence of a bottleneck effect after introduction."

The structure is clear and easy to follow.

The figures are generally clear but in Figure 3 the p-value cutoff being used should be mentioned, also it's not clear to me whether each taxon incorporates multiple OTUs or just one.

Raw data is referred to but not (yet?) available online: i.e. BioProject PRJNA524707 - but I expect this will be uploaded later?

Experimental design

Experimental design
The study presents original primary research within the scope of the journal.

The research question is clearly stated: "we tested the hypothesis that the composition of the community of fungi associated with ambrosia beetles is affected by the habitat"

The methods are generally well described but there are a few pieces of missing information:
- The old growth rests are described as "very old" I think a more rigorous age estimate is needed here.
- What percentage ethanol was used?
- What was your PCR protocol? Was it exactly the same as Kostovcik et al 2015? If so, please state this clearly.
- What quality filtering parameters did you use?
- Were OTUs clustered at 97%? Please state in text.
- What BLAST parameters did you use?
- How did you select the final ID from you blast - was it just the top hit?
- It sounds like you ran 3 separate linear mixed effects models - one for each of the alpha diversity index? If so - state this explicitly.
- While I understand that there is a reference provided for DESeq2 which was used to identify the differential presence of OTUs between different forest types, I am not familiar with the method it uses and I think the manuscript would benefit from a brief description of how this analysis is done.
- Also it's not clear to me how the significance of the alpha diversity indices were tested - a p-value is given but not test statistic or df - was it a likelihood ratio test? This needs to be stated.
- How is the percentage of explained variance obtained from the PERMANOVA?

The methods are generally rigorous but I see a few issues:
- Checking traps every 3 weeks is likely to affect the results as DNA will be degraded during this time - how this is likely to affect the results should be discussed.
- I can't see how the authors have accounted for spatial autocorrelation in their analysis, this should be done.
- Why didn't the authors use the UNITE database? Simply blasting against GenBank can lead to taxonomic inaccuracies, especially if the 'top hit' is used. I suggest using Genbank with the LCA approach in MEGAN or RDA against UNITE.
According to the discussion, the majority of sequences were not identified - the proportion of identified OTUs should be mentioned in the results and I think that this problem should be solved by using a better reference database.
- The authors used FDR to correct for multiple comparisons for the PERMANOVA but not for the linear mixed effects models? - if 3 linear mixed effects models were used this should be done.

Validity of the findings

Validity of findings
I think it's clear that there is a strong difference between the fungal community derived from X.germanus communities from old growth and new growth forests - nice NMDS. And, as stated, although X.saxesenii communities are also statistically different - this is probably just an artefact of having a lot of data points - as shown by the low level of explained variation and discussed appropriately.

It's also clear that there was a difference in the extrapolated richness (Chao1) and phylogenetic diversity between old growth and new growth forests for X.germanus where old growth growth forests have higher diversity in both cases while this was not found for X.saxesenii - this too was discussed.

However, I am not convinced by the discussion of the potential mechanisms behind these findings - i.e. the fact that habitat had a greater effect on the mycobiota of the invasive beetle species indicates "a limited co-evolution between X.germanus and environmental fungi in the invaded area or the occurrence of a bottleneck effect after its introduction". I am not really sure what the authors mean by this and think that this claim needs careful explanation with examples and references.
Additionally, there is no discussion of the 1-Simpson (dominance) results in the discussion and how these results add to the interpretation of what's happening in the system.
In lines 295-296 the authors discuss the effect of natural enemies but it is not clear what they are referring to - I assume entomopathogenic fungi that attack the beetles?
The authors also say that "it is reasonable to assume that the sudden arrival of a species in a novel environment leads to a reduced efficiency of these mechanisms." Again, I would like to see a clearer logical flow here explaining these assertions and drawing in the results of this study.

Finally, something that I really do think needs some discussion, is the fact that these fungi will come from both the gut and the mycangia (assuming the rigorous surface sterilisation did not remove the mycangial community). As such, multiple mechanisms are behind these findings. The discussion in lines 286-292 would benefit from drawing a distinction between these two sources of fungal taxa, as would the discussion in lines 318-320.

Additional comments

Other minor comments:
Line 39: Should be 'Microorganisms as an important facet'
Line 40: Should be 'Insects, like many other...'
Lines 41-42: Should be 'positive (i.e. mutualistic)' and 'negative (i.e. parasitic)'
Line 81: "and/or" should be changed to just 'and'
Line 195 - should be NMDS?

The number of specimens that were retained for analysis, along with the number of reads/OTUs etc... should be mentioned in the results.

The estimates etc... for the linear mixed effects models are not provided - I suggest that these are added to the supplementary materials.

It is mentioned that the old growth forests are "small patches". It has been shown that fragment size can alter the diversity of a community - I think the manuscript would benefit from mentioning this in the discussion.

Reviewer 2 ·

Basic reporting

I cannot say that the article was clear, unambiguous throughout:
The abstract needs to be reworked to be much more clear. There are a number of concepts in it and it must be better defined.
The introduction includes a number of important concepts and is relevant, but it is also lacking essential concepts to be defined such as;
• Molecular markers used for fungi, and their pros and cons;
• Technologies to assess communities (e.g. NGS, metabarcoding);
• Information related to old growth and restored forests (instead, it was explained it in the Materials and Method section);
• Insect traps and lures;
Unfortunately, many citations are problematic; I appreciate the authors’ effort in using recently published work but, I noticed many occurrences of “cross-referring” in the article, especially so in the introduction where base concepts were defined. For example, on line 34, the cited authors “Early et al.” are not the original authors of this information. Early et al. do talk about those facts, but the reviewed article’s authors need to cite the originator of this information (e.g. Early et al.'s citation of "Millennium Ecosystem Assessment. Ecosystems and Human Well-being: synthesis World Resources Institute (2005)". The same applies to the citation from Brockerhoff and Liebhold (2017). I will not highlight other similar "cross-referring" occurrences in this article (there are more), but they all need to be addressed, please.
* * *
More than once, I noticed that information stated sounded a bit repetitive or as if the authors were “spinning around” the idea instead of going straight to the point by using repetitive words or more words than needed. The article would benefit from fine-tuning this aspect as it would be easier to read. One example is as follow:
“In addition, exotic insects interact with native microbial communities and thus may acquire novel microorganisms in the invaded environment that may affect their invasion. Understanding the ecological factors that influence the acquisition of novel microorganisms is thus essential to clarify the mechanisms behind insect invasions.”
A statement that could possibly be shortened in something possibly more efficient such as:
“Given that exotic insects interact with native microbial communities, it is essential to better understand i) the ecological factors influencing the acquisition of new microorganisms by beetles and ii) the mechanisms involved.”
In addition, repetition of words is also occurring at times, and there are a few times where I considered the vocabulary inexact. Three examples are presented below:
Repetition:
“All these symbionts represent a main component of an insect invader as can be responsible for the success or failure of its invasion and for the impact that it can have on in the invaded ecosystem.“
Inexact vocabulary:
“In addition, exotic insects interact with native microbial communities and thus may acquire novel microorganisms in the invaded environment that may affect their invasion.”
The word "novel" is misleading as it suggests the acquisition of microorganisms that did not exist before. I suggest rephrasing with something along the lines of "never previously associated with invasive exotic insects" or "newly associated".
“The comparison between the exotic and the native species allowed us to test whether the beetle fungal communities respond equally when facing different habitats.”

What is meant by "respond"? Did the authors actually test the beetle's response or did they seldom compare communities? The term "respond" is suggesting that they looked at behavioral changes, which I don’t think they did.
* * *
Currently, it is impossible to access the raw data deposited under the NCBI SRA BioProject PRJNA524707 (stated in the article). It is essential that the data becomes available for better reviewing the article. I suspect that the release date is in the future, which prevented me from accessing it.
* * *
In general, Tables and Figures were clear, neat, and intelligible; although you will notice a few suggestions to improve some of them.

Experimental design

The introduction needs to be better structured as literature is intercalated between goals and hypotheses statements, which becomes confusing for the reader to understand the goals, research questions and their relevance. Specific examples can be found in the annotated PDF document attached.
* * *
In general, the science used appears to be sound (usage of known bioinformatics tools and statistical tests, inclusion of controls within the experiment, as well as triplicates, etc.), although given the lack of details on the Materials and Methods, it is hard for me to assess this aspect in a fair manner.
* * *
There are details missing with regards to the multi funnel traps used. It would be relevant to specify the color used, the height at which they were installed, the under vegetation type, and whether it was a wet or dry system. I believe authors used the wet system, in which case they shall add more details related to the release system for the lure. Those traps are designed so one can use ultra-high release semiochemicals that are specific to a certain group of insect.
Another concern I had is that the authors used a broad lure to attract specific beetles. Ethanol, if this is the only one that the authors used, is attractive to almost any insects and therefore I wonder if they shall have opted for a better suited semiochemical to target the desired ambrosia beetles.

I find it confusing the way it is written now, as authors mention using ethanol for baiting and then ethylene glycol for killing the insects, but very little details are given as to how are the chemicals set and released in the trap. It would be relevant to clarify how they washed the traps between each collection and what was used to do so.
* * *
A lot of details are missing in the Material and Methods, including:
• software used for data analysis;
• PCR conditions (cycling), reagents, volumes, concentration and instrument used for metabarcoding;
• Size selection for amplicon cleaning and ratio used (Ampure particles)
• Details pertaining to the processing on illumina MiSeq (e.g. was it paired-end?)
• OTU cutoff %
• Package and function used for specific analysis
Justification for each data analysis process (i.e. authors shall indicate, for example, why they ran the NMSD test.)
* * *
I noticed occurrences of misplaced sentences or statement in the “Material and Methods” section so I recommend that the article’s structure is reworked for improvement. Specific examples can be -found in the annotated PDF document attached.

Validity of the findings

When done, the explanation for part of the results performed was good and relatively clear. However, I believe it is essential that authors add more to their discussion section. More specifically, I think they shall elaborate on the pros and cons of metagenomics, OTUs, and what one shall expect from the results based on this type of analysis process. The primers used can induce biases, the OTU cutoff is quite important for proper identification, the different information extrapolated by the different diversity indices, etc.

Authors performed a decent amount of speculation to explain the differences they observed within their data and I appreciated that. I enjoyed getting an idea of their point of view on the analyses. They stated clearly that it was only speculation of course.
The conclusion could take improvement; I think that there must be a way to give it more impact and highlight better what was found. It was a little bit “plain” in my personal opinion, which is so sad to close such a load of handwork (give it what it deserves!!).

Additional comments

A few times, I requested that authors add reference to the statements they made and/or that they split sentences that are too long; for instance:

“We are aware that this approach potentially leads to amplification biais for Microascales and Ophiostomatales, which includes the main mutualists of the two ambrosia bettle species. However, in this study we did not focus on the main mutualists, which ecology is widely studied, but we targeted the entire mycobiota of X. germanus and X. saxesenii which is less described and might be important aspects of beetles’ ecology.”

On that note, I noticed the use of personal pronoun throughout the article, (i.e. “we”) and I do not recommend doing so as often as the authors did, but this might be a preference thing only.
* * *
There was sometimes lack of originality, for instance the first sentence of the abstract is about 99% identical to the first sentence of the discussion part. I noticed a lot of word repetition, among other the word “despite”.
* * *
Authors shall review the references cited; I noticed that “The R core Team” one included two different years. I did not check many more but I suspect there could be errors induced by the software used to list references (Example EndNote).
* * *
One thing that struck me was the wording “Forest continuity”: 1) because I had never heard of it, and 2) because although it is part of the title, authors barely ever used in the article. Forest continuity was described somewhat too late in the article, within the Materials and Methods, which left me wondering for a long time what was this article about exactly. Rather, authors used other “synonyms” such has “habitat characteristics” earlier in the manuscript.

Annotated reviews are not available for download in order to protect the identity of reviewers who chose to remain anonymous.

---

## Round 0.2 · Minor Revisions

Dear Dr. Rassati and colleagues:

Thank you for re-submitting your manuscript to PeerJ. I have received a second round of reviews form the two original reviewers. It appears that the reviewers are much more optimistic with your work. Excellent! However, there are still some minor issues to address. Please handle these ASAP so we may move towards accepting your work for publication.

I look forward to seeing your revision, and thanks again for submitting your work to PeerJ.

Good luck with your revision,

-joe

Reviewer 1 ·

Basic reporting

The language was clear – a few minor errors (see edited word doc).

Intro and background show the context well and the literature is well referenced and relevant.

The structure is clear and easy to follow.

The results were relevant to the hypotheses but some of the claims in the discussion were not supported by the results.

Experimental design

The study presents original primary research within the scope of the journal.

The research question is clearly stated: “In this study, we tested the hypothesis that different habitats influence the composition of the fungal community associated with ambrosia beetles, reflecting a potential acquisition of fungi from the environment.”

The methods are generally rigorous and well described but there is an important issue remaining:
BLAST isn’t an identitifaction method alone – it gives a list of hits against a database and there are a number of different ways of selecting the final identity of the sequence. See https://www.ncbi.nlm.nih.gov/books/NBK279690/
A common method is to select a cutoff (i.e. a value below which you disregard matching sequences) and this is often based on the e-value. You can then use the top hit (the matching reference sequence that has the lowest e-value) for the identity of your template sequence.
This will work for some template sequences, but often you will get multiple matching reference sequences that have an equally well matching sequence in the reference database.
In these cases you need to decide how to select the actual identity – often people use the ‘lowest common ancestor’ approach where if you have two sequences matching your template sequence with the same e-value – eg. Fusarium avenaceum and Fusarium bubigeum, your final identity would be Fusarium sp. Because the lowest common ancestor is the genus.
I would guess that what you have done is just to take the identities where there is no ambiguity – i.e. there is only one top hit as that would explain why you have such high numbers of unidentified species.

Validity of the findings

Last time I wrote: “I am not convinced by the discussion of the potential mechanisms behind these findings - i.e. the fact that habitat had a greater effect on the mycobiota of the invasive beetle species indicates "a limited co-evolution between X. germanus and environmental fungi in the invaded area or the occurrence of a bottleneck effect after its introduction". I am not really sure what the authors mean by this and think that this claim needs careful explanation with examples and references.”

I agree that the authors have clarified this point nicely now but I still think that there should be a statement in the discussion pointing out that two beetle species, one exotic, one non-exotic with no replication. While I think that the study should be published, I think it’s important not to overstate the relevance of the findings. In particular the statement in the conclusion that “we showed that the direct acquisition of microorganisms from the environment can modify the microbiome of an exotic species” seems a bit of a stretch to me.

Additional comments

In general, I think that the dataset collected for this project is really nice but that you're trying to say too much about co-evolutionary theory using data that aren’t suitable. This wasn’t clear to me in the last version as I didn’t understand the logical flow in the discussion. I think this could be easily solved by modifying the discussion to say that this study is very much a hypothesis generation exercise – that you've found evidence to suggest that this bottleneck is occurring and here's how this should be investigated further. I can see that you've already done this to an extent but the structure of the discussion is a bit disjointed so this sentiment doesn’t come across strongly enough.

Annotated reviews are not available for download in order to protect the identity of reviewers who chose to remain anonymous.

Reviewer 2 ·

Basic reporting

I can see a significant improvement in this article with regards to the ambiguities I previously raised. The manuscript is far more easy to read and I truly enjoyed it; great work. This applies as well to the abstract and I liked that the title was adjusted accordingly. I did not encounter repetitions or lenghty sentences anymore.

Authors also partially added literature background as I suggested, although they did not for two points I mentioned (i.e. technologies to assess communities (e.g. NGS, metabarcoding) and insect traps and lures) but, given their reasonable response, I will leave it to the discretion of the editor at this point.

Citations that were a concern to me (e.g. cross-citation) appear to have been adressed properly.

As previously stated, Tables and Figures were clear, neat and intelligible.

Experimental design

I appreciate the major restructuration of the introduction and I truly believe that your paper benefited a lot from that. I don't see the staggering of sections previously mentioned, which is well done. To me, it is clear and efficient as it is now. In addition, the details that I requested for were added; thank you.
Also, authors gave me satisfactory answers to my concerns with regars to insect traps, lures used, OTU cutoff and PCR settings.
While I agree that most bioinformatics details were given, I still would like to see in the Material and Mehods which function were used with the packages Vegan, phyloseq and deseq2, especially so because the function used with other packages were named later in this section. This would improve the ability to replicate the process used.
I think authors also must add the citation for USEARCH: I requested it before but it was embedded with another comment of mine, so I suspect authors may just have not seen it then.

The comment that I made in my previsous review "I believe it is essential that authors add more to their discussion section. More specifically, I think they shall elaborate on the pros and cons of metagenomics, OTUs, and what one shall expect from the results based on this type of analysis process. The primers used can induce biases, the OTU cutoff is quite important for proper identification, the different information extrapolated by the different diversity indices, etc. " still applies to this manuscript, but again, given the reasonable reply from the authors, I will leave it to the editor to cut on that.

Validity of the findings

This paper is bringing new data and is surely worth publishing.
The conclusion is still not strong enough to give this paper what it deserves, although it is currently satisfactory.

Additional comments

The sentence ending in line 319 ([...] which instead the guts.) sounds weird. I suggest rephrasing it slighly.

---

## Round 0.3 · accepted · Accept

Dear Dr. Rassati and colleagues:

Thanks for re-submitting your revised manuscript to PeerJ, and for addressing the concerns raised by the reviewers. I now believe that your manuscript is suitable for publication. Congratulations! I look forward to seeing this work in print, and I anticipate it being an important resource for research communities studying ambrosia beetle ecology and evolution.

Thanks again for choosing PeerJ to publish such important work.

-joe